# Can the Supido Radar Be Used for Measuring Ball Speed during Soccer Kicking? A Reliability and Concurrent Validity Study of a New Low-Cost Device

**DOI:** 10.3390/s22187046

**Published:** 2022-09-17

**Authors:** David M. Díez-Fernández, David Rodríguez-Rosell, Federico Gazzo, Julián Giráldez, Rodrigo Villaseca-Vicuña, Jose A. Gonzalez-Jurado

**Affiliations:** 1Department of Education, Faculty of Education Sciences, University of Almería, 04120 Almería, Spain; 2SPORT Research Group (CTS-1024), CERNEP Research Center, University of Almería, 04120 Almería, Spain; 3Research, Development, and Innovation (R&D+I) Area, Investigation in Medicine and Sport Department, Sevilla Football Club, 41005 Seville, Spain; 4Department of Sport and Informatics, Universidad Pablo de Olavide, 41013 Seville, Spain; 5Physical Performance & Sports Research Center, Universidad Pablo de Olavide, 41013 Seville, Spain; 6Faculty of Physical Activity and Sports, University of Flores (UFLO), Buenos Aires 1406, Argentina; 7Escuela de Ciencias de la Educación y Tecnología de Pedagogía en Educación Física, Universidad Católica, Silva Henríquez (UCSH), Santiago 8330225, Chile

**Keywords:** radar, agreement, concordance, physical performance, reproducibility, kicking ball speed

## Abstract

The aim was to analyze the reliability and validity of a low-cost instrument, based on a radar system, to quantify the kicking ball speed in soccer. A group of 153 male soccer players (under-13, n = 53; under-15, n = 54; under-18, n = 46) participated in this study. Each player performed three kicks on the goal in a standardized condition while the ball speed was measured with three different devices: one Radar Stalker ATS II^®^ (reference criterion) and two Supido Radar^®^ (Supido-front of the goal and Supido-back of the goal). The standard error of measurement (SEM) expressed as a coefficient of variation (CV) and the intraclass correlation coefficient (ICC) were employed for assessing the reliability of each instrument. Stalker and Supido-back showed very high absolute (CV = 4.0–5.4%) and relative (ICC = 0.945–0.958) reliability, whereas Supido-front resulted in moderate to low reliability scores (CV = 7.4–15%, ICC = 0.134–0.693). In addition, Lin’s concordance correlation coefficient (CCC) values revealed an ‘almost perfect’ agreement between Stalker and Supido-back for the average (r = 0.99) and maximal (r = 0.98) ball speed, regardless of the ball speed range analyzed. However, Supido-front resulted in a poor degree of concordance (CCC = 0.688) and a high magnitude of error (17.0–37.5 km·h^−1^) with the reference Stalker radar gun. The Supido Radar^®^ placed behind the goal could be considered a reliable and valid device for measuring ball speed in soccer.

## 1. Introduction

Soccer is considered the most popular sport in the world [1] and the analysis of soccer characteristics has been the focus of numerous scientific studies. One of the most studied skills in soccer is kicking the ball [1,2,3] because it is the main offensive action in this sport and, consequently, is considered decisive for the final result in competitions [1,4,5]. Indeed, several studies have reported that a third part of the goals scored during a match are obtained after powerfully kicking the ball from outside the area [6,7]. In addition, previous studies have indicated that the soccer teams with a greater number of kicks to the goal showed a higher probability of scoring and winning the match [8], while some authors have pointed out that the kicking ball speed can be used as a research tool to distinguish high level soccer [9] and futsal players [10] from those of lower levels.

Traditionally, the performance during the kicking ball has been assessed by measuring the maximal ball speed, as an increase in this parameter reduces the time for the goalkeeper or an opponent to intercept the ball successfully, therefore raising the probability of scoring [1,11,12]. For this reason, several studies on soccer players have focused on analyzing the kicking ball speed [8,13], its determinant factors [14,15,16], and the effect induced by different training programs [17,18]. Accordingly, due to the relevance of this variable, the use of accurate, valid, and reliable devices for measuring the kicking ball speed is warranted.

Studies analyzing the kicking ball speed in soccer players have used different devices, including high-speed cameras [4,16,19,20], photocells [13], inertial sensors [21], and radar guns [2,5,14,17,18,22,23]. In this regard, radar guns are a common tool both in practical use and scientific research due to their high accuracy [24,25]. Specifically, the hyper frequency Stalker II^®^ radar gun is the most accepted and used device and it is usually considered as reference radar to quantify the ball speed in different sports, including tennis, soccer, handball, futsal, and baseball [18,24,26,27]. A previous study analyzed the reliability of ball speed in futsal [18] and tennis players [24] using this instrument and showed a low coefficient of variation (CV < 6%) and a high intraclass correlation coefficient (ICC > 0.94). These results indicated a high absolute and relative reliability of both the participants and the measuring device. Consequently, previous studies have used the Stalker II^®^ radar gun as a reference criterion to analyze the concurrent validity of other devices for ball velocity and running speed measurements [28,29,30].

In most cases, the use of cameras, photocells, or radars to quantify the ball speed could be limited due to the high cost of these devices and the lack of practical applicability. For this reason, it is common for the use of these devices to be restricted to professional soccer teams. However, the kicking ball speed is an important motor skill for soccer players in all categories and ranges of age. Thus, this quality should be able to be measured in all soccer players. In addition to the high cost, other potential disadvantages are associated with the use of cameras, photocells, or radars for measuring the ball speed during a match or training session, including: (a) not providing the ball speed in real-time (cameras); and (b) the need for these devices to be placed on the pitch to quantify the kicking ball speed (photocells). The sizes of the instrument, durability, or autonomy are other variables limiting the actual methods to quantify the ball speed in soccer (Table 1).

Therefore, in line with Hernández-Belmonte & Sánchez-Pay [28] and providing a new alternative, it seems necessary to find accurate and reliable devices for measuring the kicking ball speed in real time, which, in addition, is not excessively expensive so that it can be used by soccer teams, regardless of their economic level. Thus, the purpose of this investigation was to analyze the reliability and concurrent validity of a small, low-cost instrument with autonomy and durability, based on a radar system, to quantify the kicking ball speed of soccer players. To the best of our knowledge, no study has yet analyzed the reliability and validity of this commercial tool.

## 2. Material and Methods

### 2.1. Participants

A group of 153 young male soccer players of three different age categories (under-13, n = 53; under-15, n = 54; under-18, n = 46) volunteered to take part in this study. The physical characteristics of the subjects are shown in Table 2. The subjects had more than 3 years of training experience, and they had been injury free for at least 4 months before participating in this study. The soccer players belonged to three different soccer clubs from Seville (Spain). The coaches and parents were informed about the different test procedures performed during the study. Parental/guardian consent for all the players under the age of 18 involved in this investigation was obtained. All of the subjects were informed about the experimental procedures and potential risks before they provided their written informed consent. The study was conducted according to the guidelines of the Declaration of Helsinki, and approved by the Ethics Committee of Pablo de Olavide University (EH-1/2015)

### 2.2. Experimental Design

This study was designed to address the following question: Is a low-cost radar system a reliable and valid device for measuring the kicking ball speed in soccer players? To investigate this question, 153 soccer players of different age categories (U13, U15, U18) kicked three shots on the goal in standardized conditions while the ball speed was measured with three different devices: two Supido Multi Sports Speed Radar^®^ (devices evaluated) and one Radar Stalker ATS II (radar gun considered as a reference criterion). All of the testing sessions were performed at the same time of day (from 6:00 to 9:00 pm) for each team and under similar environmental conditions (~28 °C and ~40% humidity).

### 2.3. Testing Procedures

Testing sessions were conducted on an artificial grass field and lasted ~50 min for each soccer team. At least three experienced researchers carried out and supervised the measurements during each testing session. During the session, each team was divided into groups of five players for performing the kicking ball test. Before the evaluation, the participants performed a general standardized warm-up consisting of 5 min of running at a self-selected intensity, 5 min of joint mobilization exercises, and three progressively faster 20-m running accelerations at 80, 90, and 100% of perceived effort, respectively. After, a specific warm-up consisting of five progressive kicks on the goal (30-s rest) and three maximum kicks on the goal (1-min rest) was performed. To improve the degree of familiarization, the soccer players performed the specific warm-up in the same conditions as the subsequent evaluation.

*Kicking ball test.* Players performed a maximal velocity kick to a stationary ball. The participants were provided with adequate clothing and footwear for the test. The kick was carried out with the dominant leg. A ball with a standard Spanish Federation of soccer size and inflation was kicked 6 m toward the goal. The players were asked to shoot the ball as hard and fast as possible. The initial distance of the player from the ball was 3 m. This distance was chosen so the subjects could make an approach race before kicking the ball. Three shots were allowed for each player, with 1-min rests between them, and the resulting mean and maximum value were kept for the subsequent analysis. If, after kicking, the ball missed the goal, the kick was not considered, and the player was required to shoot the ball again after a 1-min rest. The ball speed was measured by a reference radar gun (Stalker ATS II, Acceleration Testing System^®^) located 1 m away from the stationary ball and pointed toward the ball according to the instruction manual. In addition, the ball speed was measured by two Supido Multi Sports Speed Radar^®^ devices. One of these was located 1 m from the stationary ball and pointed toward the ball (Supido-front), right next to the Stalker ATS II radar gun. The other Supido device was located 2.2 m behind the goal (Supido-back) and also pointed towards the ball. The radar gun Stalker, with a 0.01 s precision time, a velocity range of 1–1432.3 km·h^−1^, and a capacity to identify the ball movement from 152.40 m, was the reference instrument [28,29,30] used to analyze the concurrent validity of the low-cost Supido radar device. The Stalker radar gun and the two Supido Speed Radars were used under the same conditions.

### 2.4. Statistical Analysis

Standard statistical methods were used for the calculation of means and standard deviations (SD). All variables met the assumption of normality (Kolmogorov –Smirnov test) and homoscedasticity (Levene test). A one-way repeated measure analysis of variance (ANOVA) was used to detect the differences between the three attempts of each player for each instrument [31]. To test the concurrent validity of the Supido radar with respect to the Stalker radar, a one-way ANOVA was employed to detect the differences between the three instruments [31]. Bonferroni´s post-hoc was used for the differences between the means. Absolute and relative reliability was assessed for each instrument. A one-way random effects model ICC with absolute agreement (model 2,1) was used to determine the relative reliability [32]. The size of the correlation was evaluated as follows: r < 0.7 low; 0.7 ≤ r < 0.9 moderate, and r ≥ 0.9 high [32]. Absolute reliability was reported using the standard error of measurement (SEM =$\;\sqrt[{2}]{Error\;Mean\;Square}$). The SEM values were expressed as a percentage of their respective means through the CV [31,33]. A CV of ≤10% was set as the criterion to declare a variable as reliable. The minimal difference (MD) was determined per variable using the equation: MD = SEM × 1.96 × $\sqrt[{2}]{2}$. Bland –Altman´s diagrams were used to evaluate the agreement or concordance between the instruments [34]. Pearson’s correlation coefficients were calculated to establish the respective relationships between the instruments. In addition, Lin´s concordance correlation coefficient (LCCC) was also used to evaluate the concordance between the instruments [35]. For the continuous variables, the values with this statistical indicator are classified as: ‘almost perfect’ >0.99, ‘substantial’ between 0.95 and 0.99, ‘moderate’ between 0.90 and 0.95, and ‘poor’ <0.90. Both the mean square deviation (MSD) and the variance of the difference between measurements (VMD) were used as error indicators. The closer the MSD and VMD are to zero, the better, since this indicates a constant and proportional systematic error and greater precision, respectively. Maximum errors (ME) at the 95% confidence interval were calculated from the SEE (ME_SEE_) and Bland–Altman bias (ME_BIAS_) for the different speed outcomes (m·s^−1^) analyzed. For a better analysis of the results, the data were pooled into different kicking ball speed ranges (<85 km·h^−1^, 85–99 km·h^−1^, <99 km·h^−1^). The statistical significance was set at *p* < 0.05. The main analyses were conducted with SPSS (V18.0; SPSS, Inc., Chicago, IL, USA).

## 3. Results

The descriptive data for all three kick attempts and the average of each measuring instrument mean value, according to the ball speed range, are presented in Table 3. No significant differences between the attempts were found for the Stalker and Supido-back devices. For Supido-front, there were significant differences (*p* < 0.05) for the ball speed between attempts one and two and attempt three when the data were pooled and when the ball speed was >85 km·h^−1^. Additionally, for Supido-front, a statistically significant difference was obtained between attempts one and three when the ball speed was >99 km·h^−1^. Comparisons between the instruments revealed significant differences between Supido-front and Supido-back (*p* < 0.05) and Stalker (*p* < 0.01) when the data were pooled and when the ball speed was >99 km·h^−1^ (Table 3).

When the data were pooled, Supido-back and Stalker showed a very high absolute and relative reliability (CCI: 0.945–0.958 and CV: 4.7–4.0%, Table 4). Supido-front was the instrument with the lowest absolute and relative reliability, showing higher SEM, CV, and MD scores and lower ICC scores than the other two instruments (Table 4). Furthermore, the reliability for Supido-front decreased as the speed range increased. Supido-back and Stalker presented similar reliability scores, with no substantial changes in the ICC, CV, SEM, and MD depending on the ball speed (Table 4).

The Bland–Altman analysis for the average speed values of Stalker vs. Supido-back showed a systematic bias of 0.8 ± 1.5 km·h^−1^ (maximal error: 5.8 km·h^−1^). A comparison between Stalker vs. Supido-front resulted in a systematic bias of 3.5 ± 7.7 km·h^−1^ (maximal error: 30.2 km·h^−1^). For maximum ball speed values, the Bland–Altman analysis showed a systematic bias of 0.5 ± 2.0 km·h^−1^ (maximal error: 7.8 km·h^−1^) and −0.5 ± 3.3 km·h^−1^ (maximal error: 12.9 km·h^−1^) for Stalker vs. Supido-back and Stalker vs. Supido-front, respectively (Figure 1). When the data were analyzed by ball speed ranges, the average values of Stalker vs. Supido-front showed a progressively greater systematic bias (0.1–7.5 km·h^−1^) and maximal error (17–37 km·h^1^) as the ball speed increased (Table 5 and Figure 1B), whereas the maximal error for the average ball speed values of Stalker vs. Supido-back ranged from 4.9 to 6.1 km·h^−1^ (Table 5 and Figure 1A). For maximal ball speed values, the systematic bias and maximal error for Stalker vs. Supido-back and Stalker vs. Supido-front were similar in the different ranges of ball speed analyzed (Table 6 and Figure 1C,D).

LCCC values revealed an ‘almost perfect’ agreement or concordance (r = 0.99) between Stalker and Supido-back for the mean ball speed, whereas accordance between Stalker and Supido-front was ‘poor’ (r = 0.69) for this variable (Table 5). For maximal ball speed, both Supido devices presented a ‘substantial’ agreement with respect to the Stalker device (r = 0.98 and r = 0.95 for Supido-back and Supido-front, respectively). An analysis of the ball speed ranges revealed a ‘substantial’ to ‘moderate’ concordance between Stalker and Supido-back for the mean ball speed and ‘moderate’ to ‘poor’ agreement for the maximal ball speed. The concordance between Stalker and Supido-front was ‘poor’ in all the ball speed ranges for both average and maximal speed variables (Table 6).

Correlation coefficient values between the different instruments are displayed in Figure 2. For the average values, an almost perfect correlation coefficient was observed between Stalker and Supido-back (r = 0.99, SEE = 1.48 km·h^−1^; Figure 2A), whereas the magnitude of the correlation coefficient between Stalker and Supido-front was moderate (r = 0.73, SEE = 7.13 km·h^−1^; Figure 2B). A very strong relationship was found for the maximum values between Stalker and Supido-back (r = 0.98, SEE = 1.99 km·h^−1^; Figure 2A) and Stalker and Supido-front (r = 0.95, SEE = 3.22 km·h^−1^; Figure 2B). When the data were analyzed by ball speed ranges, the correlation coefficient values between Stalker and Supido-front showed a progressively substantial decrease as the ball speed range increased for both the average and the maximal ball speed variables (Figure 2B,D). The relationship between Stalker and Supido-back also progressively decreased with the increasing of the speed range, but correlation coefficient values were always higher than 0.85 (Figure 2A,C).

## 4. Discussion

The kicking ball speed is a decisive variable used to obtain a greater probability of success, i.e., to score a goal [14,23]. For this reason, it is essential to have instruments with an acceptable reliability and validity to quantify this variable. Therefore, the aim of this study was to analyze the reliability and concurrent validity of a low-cost radar to quantify the kicking ball speed in soccer. In general, our results showed no significant differences in ball speed between attempts for any devices used, except for Supido-front in attempt three (Table 3). These results appear to indicate that the warm-up protocol used (which incorporated several maximal test attempts) was sufficient and adequate to avoid a possible motor learning effect [36,37,38], post-activation potentiation [39], or fatigue by previous kicks. In addition, Supido-front showed significantly lower average speeds than the Stalker device (reference criterion) and Supido-back, mainly at a ball speed over 85 km·h^−1^ (Table 3). In relation with this fact, it was observed that during the test, although not frequently, the Supido-front radar recorded the soccer running speed of approaching the ball, not the subsequent quantifying of the kicking ball speed. This factor could explain the lower ball speed values found in Supido-front when compared with Stalker and Supido-back. Since the instructions do not clearly specify where the Supido radar system should be placed for measuring kicking ball speed, our design aimed to check if the placement of the radar affects the ball speed measurement. These results, along with those shown in relation to reliability and concordance, seem to indicate that this device should not be placed in front of the goal to measure the kicking ball speed.

In relation to the reliability analysis, the Stalker showed very high absolute (CV = 4.0%) and relative (ICC = 0.958) reliability. These results were similar to those found in a previous study analyzing the ball speed in futsal and tennis players [18,24,28]. The Supido-back also showed high reliability scores, whereas the Supido-front showed lower ICC values and considerably higher values in the SEM, CV, and MD compared with Stalker and Supido-back (Table 4). In addition, the reliability scores for Stalker and Supido-back remained constant, regardless of the speed range analyzed. However, Supido-front presented lower reliability values as the ball speed range increased. Specifically, the MD ranged from 16.3 to 39.9 km·h^−1^, indicating a wide variability. These results contrast with the values obtained for Stalker (9.5–11.3 km·h^−1^) and Supido-back (11.1–13.5 km·h^−1^) and invalidate the use of the Supido radar in front of the goal to quantify the kicking ball speed.

When the data were pooled, the Bland–Altman analysis for the average ball speed values showed a systematic bias of 0.78 km·h^−1^ (maximum error: 5.8 km·h^−1^) and 3.53 km·h^−1^ (maximum error: 30.2 km·h^−1^) for Supido-back (Figure 1A) and Supido-front (Figure 1B), respectively. To the best of our knowledge, no previous studies have analyzed the concurrent validity of the Supido Radar^®^. However, a previous study [28] comparing the agreement between of a low-cost device (Pocket radar) and the Stalker radar (reference criterion) in soccer reported similar results to those found in our research (bias ≤ 0.83 km·h^−1^, respectively). In addition, in the same line with other low-cost radar [28], it was observed that the systematic bias and maximum error progressively increased as the ball speed ranges increased for Supido-front (Table 5 and Figure 1). The maximum error indicates the maximum ball speed difference existing between the instruments analyzed when the same kicking ball is quantified. According to our results, this value could range from 17.0 to 37.5 km·h^−1^, depending on the ball speed (Table 5), which can be considered intolerable for the measurement of this variable. These results agree with the MSD and VMD values observed for Supido-front compared to Stalker, which indicates a high non-systematic error and an important lack of precision (a greater dispersion of random error) for measuring the kicking ball speed. Thus, our results suggest that the Supido radar placed in front of the goal should not be used for quantifying the average ball speed in soccer players. According to this assertion, Pearson’s correlation analysis and Lin’s concordance correlation coefficient showed an almost perfect relationship (Figure 2A; Table 5) between Supido-back and Stalker. In contrast, Supido-front showed a poor degree of concordance with Stalker, which also progressively decreased as the ball speed range increased (Figure 2B, Table 5).

For maximal ball speed values, the Bland–Altman analysis for pooled data showed only a systematic bias of 0.47 km·h^−1^ (maximum error: 7.8 km·h^−1^) and −0.52 km·h^−1^ (maximum error: 12.9 km·h^−1^) for Supido-back (Figure 1C) and Supido-front (Figure 1D), respectively. In addition, the maximum error values for the maximal ball speed variable were lower than the average ball speed for Supido-front, although these results remained higher than those observed for Supido-back. This fact suggests that, when several attempts are recorded, the probability of Supido-front recording an attempt at a similar speed to the Stalker radar increases. However, during training or match situations, kicking the ball to the goal is carried out only once for each player, so it would be more convenient to place this device (Supido radar) behind the goal for a better and more accurate measurement of ball speed. In addition, for maximal ball speed values, both devices showed a high agreement with the Stalked radar in Pearson’s correlation analysis and Lin’s concordance correlation coefficient with a systematically greater level of concordance for Supido-back compared to Supido-front (Figure 2C,D; Table 6). This fact was more evident when the data were analyzed by ball speed ranges. While the magnitude of error remained practically constant, the degree of agreement between Stalker and Supido-front progressively decreased as the ball speed range increased so that, at ball speeds greater than 99 km·h^−1^, the degree of concordance with the Stalker radar was poor.

## 5. Conclusions

The results of the present study indicate that the position of the radar is vital and that the Supido-front showed a low absolute and relative reliability and a poor degree of concordance with the Stalker radar gun (reference criterion). In contrast, the Supido-back radar showed a high reliability and almost perfect agreement with Stalker, regardless of the ball speed range analyzed and the variable used (average or maximal speed) to quantify the ball speed, which validates the use of this instrument for kicking ball speed measurements in soccer players. Thus, coaches and researchers could obtain accurate and reliable data of kicking ball speed using a low-cost device for assessing the performance status or the effect induced after any training program in soccer players.

## Figures and Tables

**Figure 1 sensors-22-07046-f001:**
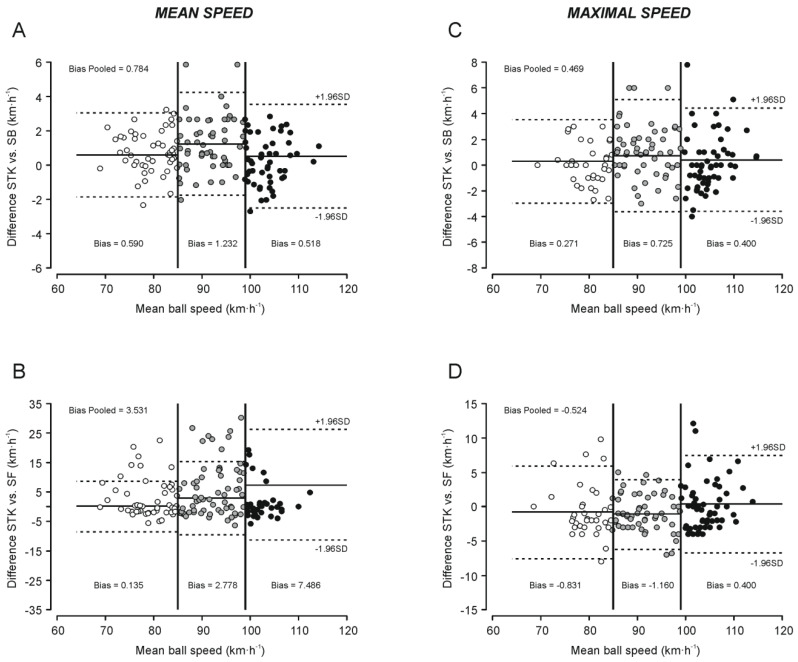
Bland–Altman plots for mean (**A**,**B**) and maximal (**C**,**D**) speed agreement analysis of the different devices used for measuring kicking ball speed. Figures show the Bland–Altman analysis for pooled data and for data divided by speed ranges. See text for details. STK: Stalker ATS II, Acceleration Testing System^®^; SB: Supido-back; SF: Supido-front.

**Figure 2 sensors-22-07046-f002:**
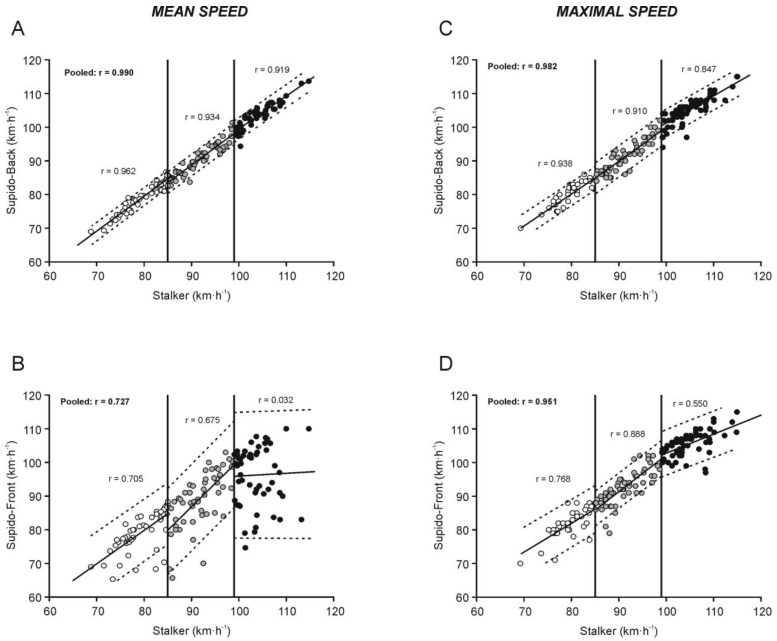
Correlation coefficient values for mean (**A**,**B**) and maximal (**C**,**D**) speed between the different devices used for measuring kicking ball speed. Figures show the Pearson’s correlation coefficient for pooled data and for data divided by speed ranges. See text for details.

**Table 1 sensors-22-07046-t001:** Technical characteristics of different devices used for measuring kicking ball speed in soccer players.

Technology	Cameras	Ultra-High-Speed Camera	Photocells	Radar Gun	Radar Low Cost
Device brand	Casio Ex-F1, ProReflex MCU1000, Casio EXZR-10, ViconMX, FKN-HC200C	MEMRECAM fx-6000	ALGE-TIMING	Doppler (Stalker Sport—Stalker ATS Pro IIs)	Supido Multi Sport Radar
Sampling frequency	200–500 Hz	5000 Hz	100–1000 Hz	24.125–34.7 GHz	2.4 GHz
Real time data	No	No	Yes	Yes	Yes
Autonomy (hands free)	Yes	Yes	No	No	Yes
Use during a match	Yes	Yes	No	Yes	Yes
Lightweight and durable	No	No	No	Yes	Yes
External power supply required	No	No	No	No	No
Price	~1000 €/1085 USD	~17000 €/18500 USD	~3600 €/4010 USD	~1500–3000 €/1600–3300 USD	~50 €/55 USD

**Table 2 sensors-22-07046-t002:** Subject’s physical characteristics (mean ± SD).

	U13 (n = 53)	U15 (n = 54)	U18 (n = 46)
Age (years)	12.6 ± 0.3	14.4 ± 0.6	16.5 ± 0.1
Height (m)	1.67 ± 0.1	1.70 ± 0.1	1.73 ± 0.1
Mass (kg)	56.3 ± 6.5	59.2 ± 7.7	65.1 ± 6.4

U13: under 13; U15: under 15; U18: under 18.

**Table 3 sensors-22-07046-t003:** Descriptive data for all three attempts and the average and maximal ball speed during the kicking ball test, according to device and speed ranges.

	Attempt One	Attempt Two	Attempt Three	Average Velocity	Maximal Velocity
Pooled (n = 153)					
Stalker	91.7 ± 10.5 ††	91.1 ± 11.1 ††	92.3 ± 11.2	92.0 ± 10.5 ††	94.9 ± 10.5
Supido-back	91.0 ± 10.8 ††	91.0 ± 11.3	91.7 ± 11.5	91.2 ± 10.6	94.5 ± 10.6
Supido-front	87.0 ± 14.0 **	87.6 ± 14.4 *	90.8 ± 12.1	88.5 ± 10.3	95.4 ± 10.3
<85 km·h^−1^ (n = 49)					
Stalker	79.9 ± 4.3	79.5 ± 5.8	79.9 ± 5.3	79.8 ± 4.3	82.7 ± 4.7
Supido-back	79.5 ± 4.7	78.7 ± 6.4	79.3 ± 5.6	79.2 ± 4.6	82.5 ± 4.7
Supido-front	79.7 ± 7.5	79.0 ± 8.6	80.2 ± 7.2	79.6 ± 6.1	83.6 ± 5.2
85–99 km·h^−1^ (n = 52)					
Stalker	91.5 ± 4.7	92.0 ± 5.3 †	92.2 ± 5.4	91.9 ± 3.9 †	95.0 ± 4.6
Supido-back	90.2 ± 5.5	90.9 ± 5.8	90.9 ± 6.2	90.7 ± 4.3	94.3 ± 5.3
Supido-front	87.2 ± 13.1 *	87.6 ± 13.3 *	92.5 ± 7.2	89.1 ± 8.2	96.2 ± 5.6
>99 km·h^−1^ (n = 52)					
Stalker	103.1 ± 4.7 †††	103.5 ± 4.7 †††	104.2 ± 4.8 ††	103.6 ± 3.7 †††	106.3 ± 3.8
Supido-back	102.7 ± 5.6 †††	102.6 ± 5.2 * †††	104.1 ± 4.8 ††	103.1 ± 3.9 †††	105.9 ± 3.6
Supido-front	93.5 ± 16.3 *	95.9 ± 15.1	99.1 ± 12.4	96.1 ± 8.9	105.9 ± 3.8

Significant differences with respect to Supido-front: † *p* <0.05; †† *p* <0.01; ††† *p* <0.001; significant differences with respect to attempt 3: * *p* <0.05; ** *p* <0.01.

**Table 4 sensors-22-07046-t004:** Reliability (ICC, SEE, CV y MD) of the different measuring devices depending on the kicking ball speed.

	ICC (CI: 95%)	SEM (km·h^−1^)	CV (%)	MD (km·h^−1^)	MD (%)
Pooled (n = 153)					
Stalker	0.958 (0.945–0.968)	3.7	4.0	10.3	11.2
Supido-back	0.945 (0.928–0.958)	4.3	4.7	12.0	13.2
Supido-front	0.634 (0.520–0.724)	10.8	12.3	30.1	34.0
<85 km·h^−1^ (n = 49)					
Stalker	0.791 (0.665–0.875)	3.4	4.3	9.5	11.9
Supido-back	0.743 (0.588–0.846)	4.0	5.0	11.1	14.0
Supido-front	0.693 (0.507–0.816)	5.9	7.4	16.3	20.5
85–99 km·h^−1^ (n = 52)					
Stalker	0.636 (0.424–0.779)	4.1	4.4	11.3	12.3
Supido-back	0.570 (0.320–0.739)	4.9	5.4	13.5	14.9
Supido-front	0.481 (0.180–0.685)	10.3	11.5	28.5	32.0
>99 km·h^−1^ (n = 52)					
Stalker	0.682 (0.497–0.807)	3.6	3.5	10.0	9.7
Supido-back	0.639 (0.430–0.781)	4.1	3.9	11.3	10.9
Supido-front	0.134 (−0.370–0.474)	14.4	15.0	39.9	41.5

ICC: intraclass correlation coefficient; CI: confidence interval; SEM: standard error of measurement; CV: coefficient of variation; MD: minimal difference.

**Table 5 sensors-22-07046-t005:** Between-device agreement (reproducibility) for the average speed during the kicking ball test according to the speed range assessed.

	MAGNITUDE OF ERROR					AGREEMENT			
	SEM (m·s^−1^)	SDC (m·s^−1^)	CV (%)	ME_SEE_	ME_BIAS_	ICC (CI 95%)	CCC [DEV (%)]	MSD	VMD
POOLED									
STK-SB	1.2	3.3	1.3	1.5	5.8	0.994 (0.991–0.995)	0.987 [1.3]	2.80	2.19
STK-SF	6.0	16.6	6.6	7.1	30.2	0.810 (0.738–0.862)	0.688 [31.2]	71.75	59.28
<85 km·h^−1^									
STK-SB	1.0	2.7	1.2	1.3	4.9	0.976 (0.957–0.986)	0.952 [4.8]	1.90	1.55
STK-SF	3.1	8.5	3.8	4.4	17.0	0.802 (0.650–0.888)	0.665 [33.5]	18.89	18.87
85–99 km·h^−1^									
STK-SB	1.4	3.8	1.5	1.5	6.0	0.940 (0.897–0.966)	0.890 [11.0]	3.87	2.35
STK-SF	4.8	13.4	5.3	6.1	24.7	0.630 (0.357–0.787)	0.477 [52.3]	47.49	39.77
>99 km·h^−1^									
STK-SB	1.1	3.2	1.1	1.6	6.1	0.953 (0.918–0.973)	0.909 [9.1]	2.66	2.39
STK-SF	8.5	23.7	8.5	9.0	37.5	−0.523 (−1.644–0.124)	0.014 [98.6]	147.55	91.55

STK: Stalker; SB: Supido-back; SF: Supido-front. SEM: standard error of measurement; SDC: smallest detectable change (sensitivity); CV: SEM expressed as a coefficient of variation; SEE: standard error of the estimate; ME_SEE_: maximum error calculated from the SEE; ME_BIAS_: maximum error calculated from the Bland–Altman bias; ICC: intraclass correlation coefficient, model; CI: confidence interval; CCC: Lin’s concordance correlation coefficient; MSD: mean square deviation; VMD: variance of the difference between measurements; Dev: percent deviation from 1.

**Table 6 sensors-22-07046-t006:** Between-device agreement (reproducibility) for the maximal speed during the kicking ball test according to the speed range assessed.

	MAGNITUDE OF ERROR					AGREEMENT			
	SEM (m·s^−1^)	SDC (m·s^−1^)	CV (%)	ME_SEE_	ME_BIAS_	ICC (CI 95%)	CCC [DEV (%)]	MSD	VMD
POOLED									
STK-SB	1.4	4.0	1.5	2.0	7.8	0.991 (0.987–0.993)	0.981 [1.9]	4.19	3.97
STK-SF	2.3	6.5	2.5	3.2	12.9	0.974 (0.964–0.981)	0.949 [5.1]	11.10	10.82
<85 km·h^−1^									
STK-SB	1.2	3.3	1.4	1.6	6.5	0.968 (0.943–0.982)	0.937 [3.3]	2.81	2.74
STK-SF	2.5	6.8	3.0	3.4	13.4	0.861 (0.755–0.922)	0.754 [24.6]	12.43	11.74
85–99 km·h^−1^									
STK-SB	1.6	4.5	1.7	2.2	8.7	0.943 (0.902–0.967)	0.891 [10.9]	5.48	4.95
STK-SF	2.0	5.5	2.1	2.6	10.2	0.920 (0.860–0.954)	0.851 [14.9]	8.08	6.74
>99 km·h^−1^									
STK-SB	1.5	4.1	1.4	1.9	8.0	0.914 (0.851–0.951)	0.840 [16.0]	4.36	4.20
STK-SF	2.6	7.1	2.4	3.2	14.2	0.712 (0.499–0.834)	0.547 [45.3]	13.27	13.11

STK: Stalker; SB: Supido-back; SF: Supido-front; SEM: standard error of measurement; SDC: smallest detectable change (sensitivity); CV: SEM expressed as a coefficient of variation; SEE: standard error of the estimate; ME_SEE_: maximum error calculated from the SEE; ME_BIAS_: maximum error calculated from the Bland–Altman bias; ICC: intraclass correlation coefficient, model; CI: confidence interval; CCC: Lin’s concordance correlation coefficient; MSD: mean square deviation; VMD: variance of the difference between measurements; Dev: percent deviation from 1.

## Data Availability

The data presented in this study are available on request from the corresponding author.

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
