# Peer review of "Can the Supido Radar Be Used for Measuring Ball Speed during Soccer Kicking? A Reliability and Concurrent Validity Study of a New Low-Cost Device"

_sensors, 2022, doi:10.3390/s22187046_

Round 1

Reviewer 1 Report

The authors presented a comparative analysis of the values of football speed measurements using the professional Stalker Sport - Stalker ATS Pro IIs and the Supido Multi Sport Radar device. The aim was to check the low-cost Supido Multi Sport Radar device for the accuracy of the measured values. This manuscript is timely. From the statistical point of view, the presented results are properly presented with reference to the literature. The discussion on the obtained results is reliable and substantive.

Nevertheless, the comments are as follows:

What is this scientific element of the manuscript, because its form is more like a research report.

The Supido Multi Sport Radar device is rather dedicated to measurements behind the goal, so the values measured in line with the ball (Supido-Front) will increase the yaw angle, which affects the accuracy of the result. What is the opinion of the authors here?

The authors focused only on the results of the analysis of the compatibility of the speed of the kicked ball only on the measurement of its speed. It would also be advisable to make a correlation on the assessment of these speeds as a function of the radar distance from the measured ball speed. In this way, a weak Supido device (Supido-Front) correlation could be quickly demonstrated.

There are no measuring characteristics of the assessed devices (Stalker Sport - Stalker ATS Pro IIs and the Supido Multi Sport Radar) in section 2.3 (Kicking ball test), i.e. in what distance range the measurements of the speed of the balls were taken.

I recommend this paper for publication after minor revision.

Reviewer 2 Report

Dear Authors,

the manuscript is interesting, well-structured, and has potential value that would benefit the community. I have a couple of concerns related to the scientific methodology. 

In particular, it should be more evident from the abstract that the first device used, i.e. the Stalker, is used as the ‘reference’ device. As stated now:

“Stalker and Supido-Back showed very high absolute (CV=4.0-5.4%) and relative (ICC=0.945-0.958) reliability…"

The reader is under the impression that the Stalker’s reliability is also validated. While this is indeed important, it is even more important to emphasize that your desire is to evaluate whether the Supido device can be used instead of the Stalker, as indicated in the title. Further on, are there any particular reasons for using these two measures, CV and ICC for validation? What about validity? Or pair-wise agreement between devices? Especially since these are considered in the and presented in the manuscript itself.

In addition, all abbreviations used in the abstract should be explained (CV, ICC).

Finally, why not consider all three kicks for later analysis, instead of considering only the mean and the maximum?

Reviewer 3 Report

Thanks to the authors for working well on this article. It is very useful because it shows the utility of using a low-cost radar which is more accessible for teams with less financial support.

Minor Edits Suggested

1. Please define the abbreviations used in the abstract

2. The differences between 1,2 and 3 are not important. Please remove these references. The differences between the measured velocities according to the stalker, supido –back and front are important. This should be highlighted and clarified.

3. Figure 1 Explain the reasoning for the different shading in the circles in the figure.

Conclusion

1. Change the first sentence to

2. The results of this study indicate that the position of the radar is vital and that the theSupido-Front showed a low absolute and relative relatability whereas the Supido-Back showed high reliability and almost perfect agreement
